# Incidence of venous thromboembolism and bleeding in patients with malignant central nervous system neoplasm: Systematic review and meta-analysis

**Viviane Cordeiro Veiga**[1]*, **Stela Verzinhasse Peres**[1], **Thatiane L. V. D. P. Ostolin**[1], **Flavia Regina Moraes**[1], **Talita Rantin Belucci**[1‡], **Carlos Afonso Clara**[2‡], **Alexandre Biasi Cavalcanti**[3‡], **Feres Eduardo Aparecido Chaddad-Neto**[4‡], **Gabriel N. de Rezende Batistella**[3‡], **Iuri Santana Neville**[5‡], **Alex M. Baeta**[1‡], **Camilla Akemi Felizardo Yamada**[1], on behalf of the TROMBOGLIO Study Group¶

**1** BP–A Beneficência Portuguesa de São Paulo, São Paulo, Brasil, **2** Hospital do Câncer de Barretos, São Paulo, Brasil, **3** HCor–Research Institute, São Paulo, Brasil, **4** Universidade Federal de São Paulo, São Paulo, Brasil, **5** Instituto do Câncer de São Paulo, São Paulo, Brasil

☯ These authors contributed equally to this work.
‡ TRB, CAC, ABC, ISN and AMB also contributed equally to this work.
¶ Membership of the TROMBOGLIO Study Group is provided in the Acknowledgments.
* viviane.veiga@bp.org.br

**Data Availability Statement:** All relevant data are in the manuscript and its supporting information files.

## Abstract

### Purpose

Central nervous system (CNS) malignant neoplasms may lead to venous thromboembolism (VTE) and bleeding, which result in rehospitalization, morbidity and mortality. We aimed to assess the incidence of VTE and bleeding in this population. Methods: This systematic review and meta-analysis (PROSPERO CRD42023423949) were based on a standardized search of PubMed, Virtual Health Library and Cochrane (n = 1653) in July 2023. After duplicate removal, data screening and collection were conducted by independent reviewers. The combined rates and 95% confidence intervals for the incidence of VTE and bleeding were calculated using the random effects model with double arcsine transformation. Subgroup analyses were performed based on sex, age, income, and type of tumor. Heterogeneity was calculated using Cochran's Q test and I² statistics. Egger's test and funnel graphs were used to assess publication bias. Results: Only 36 studies were included, mainly retrospective cohorts (n = 30, 83.3%) from North America (n = 20). Most studies included were published in high-income countries. The sample size of studies varied between 34 and 21,384 adult patients, mostly based on gliomas (n = 30,045). For overall malignant primary CNS neoplasm, the pooled incidence was 13.68% (95%CI 9.79; 18.79) and 11.60% (95%CI 6.16; 18.41) for VTE and bleeding, respectively. The subgroup with elderly people aged 60 or over had the highest incidence of VTE (32.27% - 95%CI 14.40;53.31). The studies presented few biases, being mostly high quality. Despite some variability among the studies, we observed consistent results by performing sensitivity analysis, which highlight the robustness of our findings. Conclusions: Our study showed variability in the pooled incidence for

**Funding:** VCV, SVP, TLVDPO, FRM, TRB, CAC, ABC, FECN, GNRB, ISNR, AMB and CAFY. The study is being developed at BP - A Hospital Beneficência Portuguesa of São Paulo with support of Brazilian Ministry of Health, project number NUP 25000.1121542022-98 PROADI / SUS. URL: https://www.in.gov.br/web/dou/-/extrato-de-ajuste-520387500 The authors declare no conflict of interest. The funders had no role in the design of the study; in the collection, analyses, or interpretation of data; in the writing of the manuscript; or in the decision to publish the results.

**Competing interests:** The authors have declared that no competing interests exist.

both overall events and subgroup analyses. It was highlighted that individuals over 60 years old or diagnosed with GBM had a higher pooled incidence of VTE among those with overall CNS malignancies. It is important to note that the results of this meta-analysis refer mainly to studies carried out in high-income countries. This highlights the need for additional research in Latin America, and low- and middle-income countries.

## Introduction

Central nervous system (CNS) tumors are among the ten most common types of cancer in middle-aged adults, especially in women [1]. Malignant primary CNS tumors represent 1 to 2% of all cancers in adults [2], with glioblastoma multiforme (GBM) accounting for 49% of this group [3]. Global demographic and epidemiological trends indicate an increase in the incidence in coming decades [4], especially in low and medium-income countries [5]. The main risk factors are related to family history, age, male sex, human immunodeficiency virus (HIV) infections, ionizing radiation exposure, pesticides, and cyclic aromatic hydrocarbons [6].

Among the outcomes related to primary CNS tumors, venous thromboembolism (VTE) is one of the main causes of rehospitalization and increased morbidity and mortality [4, 7]. Among tumors, CNS neoplasm is related to the greater incidence of annual thrombotic events (200 per 1,000 person-years) [8], most occurring from three to six months after diagnosis, associated with early mortality. The etiology of these events is multifactorial and includes venous stasis, direct activation of the coagulation cascade due to tissue damage, as well as pro-coagulation effects specific to the tumor [9]. Other risk factors include advanced age, tumor size, steroid use, chemotherapy, and radiotherapy [9]. Among CNS tumors, GBM with wild type isocitrate dehydrogenase (IDH) has a poorer prognosis and higher incidence of thrombotic events, estimated at approximately 20–30% per year [9].

The increased risk of venous thromboembolism (VTE) in cancer patients is particularly notable, and in neurosurgery, the introduction of pharmacological prophylaxis is well established in the literature and should be instituted 24 hours after the procedure [10, 11]. Given the substantial pathogenic propensity inherent to central nervous system (CNS) neoplasms, triggering thrombotic events such as ischemic stroke, myocardial infarction, peripheral arterial disease, and deep vein thrombosis (DVT), anticoagulants are a preventive measure against these potential risks. Conversely, the occurrence of minor or major hemorrhagic events in internal organs is closely connected to the underlying pathological condition of the disease and may be exacerbated by the prophylactic or therapeutic administration of anticoagulants. In this clinical milieu, it is imperative to meticulously assess and balance the inherent risk associated with these two complications. The aim of this systematic review and meta-analysis was to assess the incidence of VTE and bleeding in adults diagnosed with malignant primary CNS neoplasm.

## Materials and methods

This systematic review and meta-analysis were developed and conducted according to the recommendations in the manual of the Joanna Briggs Institute, and Preferred Reporting Items for Systematic Reviews and Meta-Analyses (PRISMA) [12, 13]. The protocol of this review was previously developed and registered on the International Prospective Register of Systematic Reviews Platform (PROSPERO CRD42023423949).

The scope of this review was established based on Condition, Context and Population (CoCoPop), i.e., Condition (VTE and bleeding), Context (post-diagnosis or postoperative, regardless active treatment and prophylaxis), and Population (adults diagnosed with malignant primary CNS neoplasm).

They used the following definition for VTE: any symptomatic or incidental event involving the upper or lower limbs, confirmed by imaging examinations such as venous Doppler ultrasound and/or computerized tomography of the lungs, lung scintigraphy, and angiography [7]. Arterial thromboembolic events and splenic vein thrombosis were excluded. Bleeding was defined as a fatal or symptomatic hemorrhage in a critical area or organ (intracranial, intraspinal, intraocular, retroperitoneal, pericardial, non-operated or intramuscular joint with compartment syndrome) [14].

## Eligibility criteria and outcomes

The following were considered eligible: (1) cohort studies (prospective and retrospective), case-control nested cohort studies, and cohort nested case-control studies that (2) assessed the presence of VTE and bleeding in (3) patients with malignant primary CNS neoplasm. The studies were deemed eligible when presenting, at least, a numerator and denominator for the total sample to calculate the event of interest. Review and metanalysis studies, letters to the editor, opinion articles, comments, short communications, ecological studies, and abstracts published in the annals of scientific events were not included. Studies that investigated metastatic tumors, CNS lymphoma and meningioma below grade 3 were also excluded.

The study search was limited to the period between 2013 and 2023. The period was limited to the largest number of publications on the topic and with more homogeneous patient selection criteria. Only studies published in English, Portuguese and Spanish were analyzed given the familiarity of the reviewers with these languages.

## Search strategy and screening

Potentially eligible citations were identified through a search of PubMed, Virtual Health Library and Cochrane databases. Two reviewers (SVP, TLVDPO) developed the search strategy based on the Health Science Descriptors/Medical Subject Headings (DeCS/MeSH), combining descriptors, entry terms, and free vocabulary, if needed. Specialists in the area, physicians with a specialization in intensive care and neuro-oncologists, and oncology nurses were consulted to ensure the comprehensiveness and sensitivity of the search (VCV, CAFY, and FRdM, respectively). Although standardized, the search strategy underwent minimum adjustments (i.e., minor differences in filters, for instance, using Full text, Case Reports, and Observational Studies, in the last 10 years, in English, Portuguese, Spanish, in Adults 19 years or older. Excluded were preprints as filters for PubMed, while fulltext, type_of_study:"observational_studies" OR "incidence_studies" OR "prevalence_studies", la:"en" OR "es" OR "pt", and year cluster for BVS) according to the database investigated as shown in the supplementary file (S1 Table. Search strategy according to electronic databases). However, the search focused on MeSH terms for all databases since PubMed, BVS and Cochrane considered them controlled vocabulary. The search was structured around three main concepts (central nervous system tumor, thrombosis, and bleeding) and described according to the databases.

The search results were saved in txt, ris or csv formats. Then, the files were saved in Rayyan® software to screening and eligibility. A priori, possible duplicate citations were identified and manually verified by the reviewers. Two independent blind reviewers (SVP and TLVDPO) analyzed the titles and abstracts. In case of disagreement, a third reviewer (FRdM) analyzed the studies. After study selection, eligibility was determined by retrieving the studies,

and three reviewers (SVP, FRdM and TLVDPO) analyzed the full texts, under the supervision and guidance of specialists from the area (VCV, CAFY). The reasons for exclusion were listed in the two stages, according to the eligibility criteria established (Study type, Population, Outcome, Language, Unavailable and/or Not retrieved).

## Data collection and synthesis

Specialists (VCV, CAFY, FRdM) were consulted to ensure study eligibility. Two independent blind reviewers (SVP, TLVDPO) extracted the data using a previously developed standardized spreadsheet. The extracted data were divided into (1) overall study characterization (citation, year and country of publication, study design, study period, follow-up) and (2) participants and main outcomes (total sample, sex, age and/or age range, tumor type and/or site, treatment, and health-related events). This information was extracted using REDCap®, which was previously structured. Both screening and data extraction were conducted in accordance with the guide produced during development of the study protocol. In addition, funding sources or possible conflicts of interest were registered, when available in the studies included. The main findings were presented in tables and figures accompanied by a narrative synthesis.

## Risk of bias

The studies were critically assessed by two independent reviewers (SVP, TLVDPO) using the Prevalence Critical Appraisal Tool proposed by the Joanna Briggs Institute [12]. A third reviewer assessed the studies in case of any disagreement. The studies were evaluated based on the number of positive responses (i.e., No, Unclear and Yes) and overall analysis (i.e., Include, Exclude, or Seek more information).

## Statistical analysis

Descriptive analysis of the data was carried out using absolute and relative frequencies. Meta-analysis results were developed independently for VTE and bleeding as outcomes. The RStudio program (version 4.5, metaprop package) was used for data analysis. The combined rates and 95% confidence intervals (CI) for the incidence of VTE and bleeding were calculated using the random (DerSimonian-Laird) effects model with double arcsine transformation. Subgroup analyses were performed based on sex (e.g., female and male), age ($\geq$ 60 years old), income (high income and upper-middle income countries), and type of tumor (overall CNS and GBM). The random effects model was used rather than its fixed effects counterpart due to the heterogeneity of the studies. Heterogeneity was calculated using Cochran's Q test and $I^2$ statistics (i.e., quantifying the proportion of total variation between the studies). The $I^2$ values were classified as insignificant (0–25%), low (26–50%), moderate (51–75%) and high heterogeneity ($>$ 75%). Egger's test and funnel graphs were used to assess publication bias. Significance was set at 5% for all the tests. Finally, sensitivity analyses were performed for subgroups, specifically focusing on the overall CNS and GBM. Each analysis was reiterated by excluding studies with a high risk of bias and outliers' values.

## Protocol deviations

Some protocol deviations must be addressed. The protocol did not contemplate the use of filters in the databases and did not include publication date restrictions. However, given the broaden scope adopted, mainly regarding of tumor types and events of interest, the use of filters favored study screening. Thus, the filters used were described in the search strategy according to the databases, including the data of publication (less than or equal to 10 years),

language (Portuguese, English, Spanish), study type/design and others (i.e., exclusion of pre-prints, adults over 19 years of age and full texts), when available. Descriptors and entry terms for study type/design and analysis of interest in this review were also used (e.g., prevalence, incidence, hazard ratio, odds ratio). In addition, we only searched in PubMed, BVS and Cochrane. Other databases included in our protocol, such as Embase, contain a high rate of duplicate citations with PubMed and Cochrane. We prioritized BVS because it includes Latin American and Caribbean publications. When considering medRxiv, Google Scholar and OpenGrey, the broad scope of the search strategy required a focus on better-quality evidence in order to ensure data reliability. The subgroup analysis was adjusted to better describe the findings. Based on the data availability in the evaluated studies, subgroup analysis was carried out according to sex and age. Even though it was not considered a priori, a sensitivity analysis was conducted by removing the outliers from the sample, which could be related to the study design and their methodological quality.

## Results

Database searches were carried from June to July 2023. A total of 1653 potentially eligible citations were identified (Fig 1). Possible duplicate studies were identified by the Rayyan® software, but reviewers verified one by one and resolved manually. A priori, 1589 titles and abstracts were screened. Among the excluded studies, the primarily reason was the population. The studies excluded were listed with the reasons. Studies that included different cancer types and sites were not considered eligible. Only 36 studies were eligible and included in the present review[15–50], 28 showed information for VTE and 18 for bleeding. The study screening process was summarized in a flowchart, as can be seen in Fig 1.

Overall, the studies were retrospective cohorts (n = 30, 83.3%) published in North America (n = 20), Europe (n = 9) and East Asia (n = 7). The United States (n = 18, 50%), China (n = 5, 13.8%) and Germany (n = 4, 11.1%) were the countries that published most of the included studies. Most studies included were published in high-income countries, except for 5 (13.8%) from countries classified as upper-middle income (Table 1). Neither lower-middle or low income appeared among the countries of publication from the analyzed studies. Among the studies examined in Table 1, noteworthy findings pertain to the risk factors identified in both univariate and multivariate analyses. Across these studies, advanced age was correlated with the outcomes under investigation in this meta-analysis. Performance status was similarly highlighted in three studies, while a history of VTE was documented in two. Additional studies were specifically chosen to illustrate outcome incidence, although the primary focus on risk factors, along with their corresponding adjustment variables, was directed toward the outcomes of death or recurrence.

### Participants

The sample size of studies varied between 23 and 21,384 adult patients. In studies that provided incidence values by sex (n = 9), 62.7% of participants were men (n = 1,779). Only three studies presented data on patients aged 60 years or older (28,31–32), totaling 163 patients with VTE out of 446 patients with CNS (36.5%).

Of the 36 studies included, 20 were based on gliomas (n = 30,045). It is noteworthy the study by Missios et al. [37] that analyzed a sample of 21,384 gliomas. Regarding patients diagnosed with GBM, 19 studies were identified, with a total sample size of 8,390.

When taken into account the outcomes, we highlighted whether the studies analyzed only one of the outcomes or both of them. The number of events for each outcome can be seen in Table 1 that also includes a general patients' characteristics (type of cancer, total number of

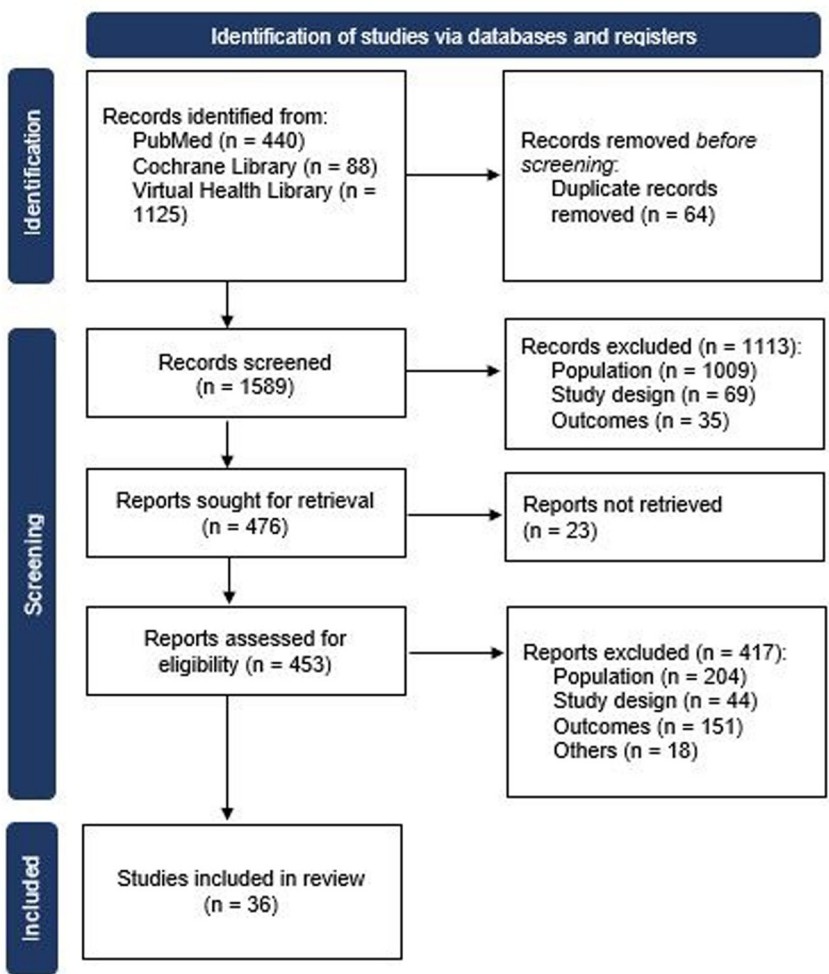

**Fig 1. Flowchart showing selection and inclusion of observational studies in systematic review and meta-analysis.**

patients, proportion of participants from sex and age) and the study period. Unfortunately, other cancer- and patient-related aspects were not included due to a heterogenous patients' characterization among the studies.

### Incidence of VTE and bleeding

Our findings were described separately for VTE and bleeding. In general, the pooled incidence ranged from 1.48 to 50.51% for VTE and from 1.69 to 45.86% for bleeding.

For overall malignant primary CNS neoplasm, the pooled incidence VTE was 13.68% (95% CI 9.79; 18.79). Similarly, the pooled incidence for bleeding was 11.60% (95%CI 6.16; 18.41).

To assess outcomes in specific subpopulations within the set of studies, subgroup analysis was performed. The studies were grouped based on specific characteristics, such as sex, age group, type of tumor, and country income. When only GBM were considered, the pooled incidence was 16.10% (95%CI 10.52; 22.57) for VTE and 8.29% (95%CI 3.26; 15.24) for bleeding. Except for GBM, the others subgroup analyzes were summarized in Table 2 (Fig 2). The forest plots for subgroup analysis can be found in the supplementary material (S1A-S1F Fig in S1 File).

**Table 1. Patients' characteristics of the included studies.**

| Author | Type of cancer | Sample Size‡ | Sex (M/F) | Age | VTE | Bleeding | Study period | Cohort | Variable as Risk factors |
|---|---|---|---|---|---|---|---|---|---|
| Auer et al., 2017 | GBM | 82 | 56/26 | 56.5 (28–78) | NR | 3 | 2006–2014 | R | No evidence was observed |
| Barbaro et al., 2022 | HGG | 152 | 96/57 | 61.5 (21–87) | 4 | 5 | 2014–2019 | R | NA (death as outcome) |
| Bruhns et al., 2018 | GBM | 71 | 45/26 | 59 | NR | 19 | 2011–2016 | R | NA (death as outcome) |
| Carney et al., 2018 | Glioma, Astrocytoma and Ependymoma | 67 | 41/26 | 56 (26–89) | NR | 9 | 2011–2018 | R | No evidence was observed |
| Diaz et al., 2021 | Glioma | 480 | Grade II: 49/75 Grade III: 59/50 Grade IV: 194/140 | Grade II: 42.7 (32.4–53.2) Grade III: 50.0 (35.2–59.6) Grade IV: 61.3 (52.6–70.4) | Grade II: 12 Grade III: 10 Grade IV: 103 | NR | 2005–2017 | R | IDH wild-type |
| Ebeling et al., 2018 | GBM | 153 | Non-IPC: 44/34 IPC: 47/28 | Non-IPC: 50.7 (26–75) IPC: 53.1 (24–76) | 12 | NR | 2009–2015 | R | No evidence was observed |
| Eisele et al., 2021 | GBM | 414 | Non-VTE: 216/133 VTE: 45/20 | Non-VTE: 63.0 (18–90) VTE: 59.7 (37–83) | 65 | 14 | 2005–2014 | R | For VTE: History of a prior VTE |
| Ening et al., 2014 | GBM | 233 | Without complication: 39/35 With complication: 78/81 | Without complication: 57.9 With complication: 63.9 | NR | 7 | 2006–2011 | R | Older Age, Radiotherapy, Chemotherapy, performance status, comorbidities, eloquent tumor location, |
| Fisher et al., 2014 | GBM and Other CNS malignancy | 2,424 | NR | NR | 670 | 290 | 1993–2006 | R* | NA (comorbidities as outcome) |
| Helmi et al., 2019 | GBM | 163 | No DVST: 98/53 DVST: 9/3 | No DVST: 54.0 ± 10.3 DVST: 51.6 ± 9.2 | 12 | NR | 2009–2015 | R | Tumor invasion of dural sinuses and greater T1/fluid-attenuated inversion recovery ratios |
| Huang et al., 2022 | GBM | 131 | 80/51 | 63 (58–67) | 48 | NR | 2017–2019 | P | Performance status, D-dimer and EGFR amplification Status |
| Jo et al., 2022 | HGG | 220 | 120/100 | LMWH: 58 (21–84) Non-AC with VTE: 61 (41–85) Non-VTE: 59 (21–85) | 22 | 43 | 2005–2016 | R | Bleeding as outcome and VTE as independent variable. No evidence was observed |
| Kaptein et al., 2021 | GBM | 967 | 580/387 | 63 (12) | 101 | 126 | 2004–2020 | R | For VTE: Older age, type of surgery, and performance status. For bleeding: VTE. |
| Kaye et al., 2023 | GBM | 293 | EGFR Non-Amplified: EGFR-Amplified: | EGFR Non-Amplified:64 (17–95) EGFR- Amplified: 64 (35–84) | 148 | NR | 2015–2021 | R | EGFR (not-amplified for sub-groups age > 60) |
| Khoury et al., 2016 | GBM | 523 | 107/66 | 65 (34–89) | 173 | 17 | 2007–2013 | R | No evidence was observed |
| Lee et al., 2019 | HGG | 918 | VTE: 66/33 Control: 537/282 | VTE: 56.6 (10.4) Control: 56.3 (8.1) | 99 | NR | 2009–2015 | R* | No evidence was observed |
| Lee et al., 2022 | HGG | 203 | 115/88 | 54 (19–76) | 3 | 5 | 2015–2020 | R | NA (death and progression-free as outcome) |

(*Continued*)

**Table 1.** (Continued)

| Author | Type of cancer | Sample Size‡ | Sex (M/F) | Age | VTE | Bleeding | Study period | Cohort | Variable as Risk factors |
|---|---|---|---|---|---|---|---|---|---|
| Lim et al., 2018 | GBM | 115 | 75/40 | 57 (23–83) | 23 | NR | 2010–2014 | R | NA (death as outcome) |
| Liu et al., 2019 | GBM | 404 | 257/147 | 59 (20–91) | 14 | 14 | 2010–2014 | R | For VET: Preoperative status performance; For bleeding: Postoperative Arterial pressure fluctuation |
| Liu et al., 2023 | Gliomas, Glio-neuronal and neuronal tumors, Anaplastic meningioma/ ependymomas | 456 | 284/172 | 56 (46–66) | 84 | NR | 2018–2021 | R | Age $\geq$ 60, preoperative abnormal APTT, operation duration longer than 5 h, admission to ICU, intraoperative plasma transfusion |
| Mantia et al., 2017 | GBM, Anaplastic oligodendroglioma, Anaplastic astrocytoma | 133 | Enoxaparin: 33/17 Control: 48/45 | Enoxaparin: 62 (26–89) Control: 61 (24–82) | NR | 61 | 2000–2016 | R¥ | Platelets, albumin, no congestive heart failure, warfarin, age, race, diastolic blood pressure, stroke |
| McGahan et al., 2017 | GBM | 39 | Gbm with hemorrhage: 8/9 GBM without hemorrhage: 12/10 | GBM with hemorrhage: 68.2 (30–71) GBM without hemorrhage: 60.6 (35–84) | NR | 17 | 2007–2013 | R | higher IHC staining for CD34 and CD105. |
| Missios al., 2015 | Glioma | 21,384 | 8924/12260 | 53.99 ± 15.91 | 788 | NR | 2005–2011 | R | Older Age, gender, West region hospitals, cardiovascular disease, coagulopathy, length of stay, seizures. |
| Nakano et al., 2018 | LGG, HGG, | 23 | NR for specific subgroup | NR for specific subgroup | 7 | NR | 2014–2017 | R | Infection |
| Nazari et al., 2020 | Glioma | 193 | 121/72 | 55 (44–66) | 26 | NR | 2003–2014 | P | Circulating lymphocytes |
| Park et al., 2021 | GBM, Anaplastic astrocytoma, Anaplastic oligoastrocytoma, Medulloblastoma | 34 | 12/22 | 60.7 (55.3–66.1) | NR | 9 | 1999–2021 | R | No evidence was observed |
| Rahman et al., 2015 | GBM | 196 | 119/77 | 59 (23–90) | 31 | 11 | 2006–2010 | R | NA (death as outcome) |
| Rinaldo et al., 2019 | Glial-based tumor | 784 | NR for specific subgroup | NR for specific subgroup | 10 | NR | 2012–2017 | R | For VTE: Age, History of VTE, Pre- or postop motor deficit, Postop intracranial hemorrhagic, Intubated >24 hrs/reintubated |
| Seidel et al., 2013 | Glioma | 3,889 | NR | NR | 143 | 123 | 2004–2010 | P | No risk analysis |
| Senders et al., 2018 | HGG | 301 | 176/125 | 57.7 ± 13.2 | 20 | 9 | 2007–2013 | R¥ | For VTE: immobility and high body mass index. For bleeding: prolonged thromboprophylaxis. |
| Shi et al., 2020 | LGG, HGG, Glioneuronal | 492 | NR for specific subgroup | NR for specific subgroup | 73 | NR | 2018–2019 | R | Older age, BMI, preoperative APTT, D-dimer, tumor histology, and surgery duration |
| Streiff et al., 2015 | HGG | 107 | 52/55 | 57 (28–85) | 26 | NR | 2005–2008 | P | Patients without complete resection and high factor VIII activity |
| Thaler et al., 2013 | Gliomas | 82 | NR for specific subgroup | NR for specific subgroup | 13 | NR | 2003–2010 | P | No evidence was observed |

(*Continued*)

**Table 1.** (Continued)

| Author | Type of cancer | Sample Size‡ | Sex (M/F) | Age | VTE | Bleeding | Study period | Cohort | Variable as Risk factors |
|---|---|---|---|---|---|---|---|---|---|
| Unruh et al., 2016 | Glioma | 317 | Discovery Cohort IDH1/2 Wild-type: 61/56 IHD1/2 Mutant: 27/25 Validation Cohort IDH1/2 Wild-type: 67/47 IHD1/2 Mutant: 20/14 | Discovery Cohort IDH1/2 Wild-type: 60.6 ± 1.1 IHD1/2 Mutant: 39.4 ± 1.6 Validation Cohort IDH1/2 Wild-type: 64.4 ± 1.3 IHD1/2 Mutant: 46.0 ± 2.1 | 61 | NR | 2009–2014 | P | IDH1 wild-type |
| Zhang et al., 2023 | Glioma | 435 | Non-VTE: 204/150 VTE: 53/28 | Non-VTE: 42 VTE: 55 | 81 | NR | 2012–2021 | R | Age, operation time, systemic immune-inflammation index (SII) and hypertension. |
| Zhou et al., 2022 | HGG | 154 | Recurrence: 38/27 Nonrecurrence: 54/35 | Recurrence: 48.95 ± 13.00 Nonrecurrence: 49.10 ± 11.90 | NR | 48 | 2016–2021 | R | NA (recurrence as outcome) |

‡ values presented for valid malignancy; DVST: Dural Venous Sinus Thrombosis; DVT: Deep Vein Thrombosis; EGFR: Epidermal Growth Factor Receptor; F: Females; GBM: Glioblastoma Multiforme; HGG: High Grade Glioma; IDH: Isocitrate Dehydrogenase; LGG: Low Grade Glioma; LMWH: Low-molecular-weight heparin; M: Males; NA: Not applicable; NR: Not reported; P: Prospective; R: Retrospective; VTE: Venous Thromboembolism; * case-control nested cohort; ¥ cohort nested case-control.

When considering upper-middle income countries, the pooled incidence of VTE was 12.68% (95% CI 3.43; 26.13) with an $I^2$ of 82.0%. This value indicated lower heterogeneity than found for overall analysis ($I^2$ = 99%). We also highlighted the wider confidence interval in addition to the fact that this subgroup consisted of only three studies. Among these studies, none reported bleeding events. For high income countries, the pooled incidence for both outcomes did not vary when considering the confidence intervals and the heterogeneity values (as can be seen in Table 2).

Among the demographic characteristics analyzed, the subgroup with elderly people aged 60 or over had the highest incidence of VTE (32.27% - CI95% 14.40;53.31) as expectedly.

According to sex, we observed a decrease in heterogeneity that reached 88% for men and 81% for women with pooled incidence of 16.52% and 15.46% respectively.

## Assessment of publication bias

In the visual analysis through the funnel plot, we observed the presence of asymmetry in the studies as can be seen in Fig 3, where studies with larger sample sizes cluster at the top but extend beyond the confidence interval. Egger's test was applied for analysis with ten or more studies due to methodological limitations.

In overall tumor analysis, the asymmetry was identified for VTE and bleeding (p < 0.001 and p = 0.011, respectively). Regarding high income countries subgroup, there was asymmetry for both VTE (p = 0.001) and bleeding (p = 0.009) (Table 2). Conversely, the studies on GBM tumors did not showed asymmetry in both analyzes for VTE (p = 0.216) and bleeding (p = 0.303). These analyzes can be found in Supporting information.

**Table 2. Summary of VTE and bleeding pooled incidence according to overall and subgroup sample.**

| Number of studies (Figure) | Group | Events | Pooled Incidence (95%CI)$^{¥}$ | $I^2$ | $t^2$ | p* |
|---|---|---|---|---|---|---|
| 28 studies (Fig 2A) | CNS | VTE | 13.68 (9.79; 18.09) | 99% | 0.0249 | <0.001 |
| 18 studies (Fig 2B) | CNS | Bleeding | 11.60 (6.16; 18.41) | 97% | 0.0398 | 0.011 |
| **Number of studies (Figure)** | **Subgroup** | **Events** | **Pooled Incidence (95%CI)$^{¥}$** | $I^2$ | $t^2$ | **p*** |
| 16 studies (Fig 2C) | GBM | VTE | 16.10 (10.52; 22.57) | 98% | 0.0264 | 0.216 |
| 10 studies (Fig 2D) | GBM | Bleeding | 8.29 (3.26; 15.24) | 95% | 0.0284 | 0.303 |
| 8 studies (S1a Fig in S1 File) | CNS Sex = male | VTE | 16.52 (10.25; 23.89) | 88% | 0.0152 | NA |
| 8 studies (S1b Fig in S1 File) | CNS Sex = female | VTE | 15.42 (9.41; 22.63) | 81% | 0.0137 | NA |
| 3 studies (S1c Fig in S1 File) | CNS Age ≥ 60 | VTE | 32.27 (14.40; 53.31) | 95% | 0.0355 | NA |
| 3 studies (S1d Fig in S1 File) | CNS Upper-middle income countries | VTE | 12.68 (3.43; 26.13) | 83% | 0.0190 | NA |
| 25 studies (S1e Fig in S1 File) | CNS High income countries | VTE | 13.28 (9.03;18.19) | 99% | 0.0278 | 0.001 |
| 17 studies (S1d Fig in S1 File) | CNS High income countries | Bleeding | 12.29 (6.50; 19.53) | 97% | 0.0405 | 0.009 |

$^{¥}$ Random effect model analysis

* p value calculated by using Egger test, NA = Not applicable (Egger test was not conducted due to total number of studies minor than 10); CI: Confidence interval; CNS: central nervous system; GBM: glioblastoma multiforme; VTE: venous thromboembolism.

## Risk of bias

The studies were assessed by two reviewers. The results were presented by study and general summary (Fig 4). The studies presented few biases, especially regarding the methods used for identification of the outcomes, being mostly high quality. Among the strengths, adequate sample size and its relationship with the analyses conducted were prominent.

Based on these findings, a sensitivity analysis was performed. Studies that received, at least, one response 'No' on any item were excluded. Accordingly, case-control studies, which have the lowest level of evidence among the selected study types, were also excluded as previously proposed in the protocol. Finally, studies with outlier incidences were identified for exclusion.

Despite the variability found among the studies, we observed consistent results considering the performed for each step of sensitivity analysis, which highlight the robustness of our findings, with outlier analysis being conducted. The summary of sensitivity analysis can be seen in Table 3. (S2 Forest and funnel plots of venous thromboembolism and bleeding according sensitivity analysis, and 95%CI graph for sensitivity analysis groups (S2A Fig).

## Discussion

We found a pooled incidence of 13.68% and 11.60% for VTE and bleeding, respectively, in adults with malignant CNS neoplasm. The occurrence of VTE in the first six months post-diagnosis is up to 7-fold higher in patients with brain tumors [51]. Previous systematic reviews already investigated the relationship between cancer and having a VTE. The majority of them assessed the risk for these events and commonly included several types and sites of cancer,

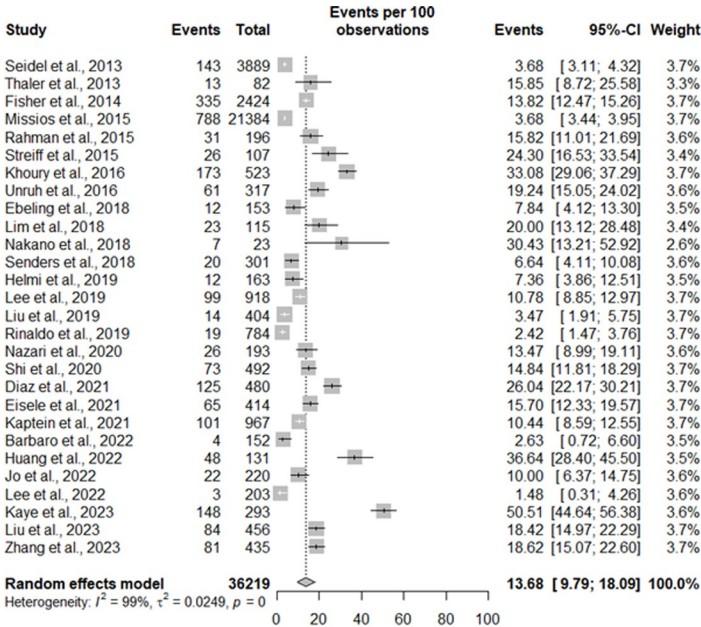

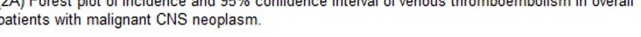

(2A) Forest plot of incidence and 95% confidence interval of venous thromboembolism in overall patients with malignant CNS neoplasm.

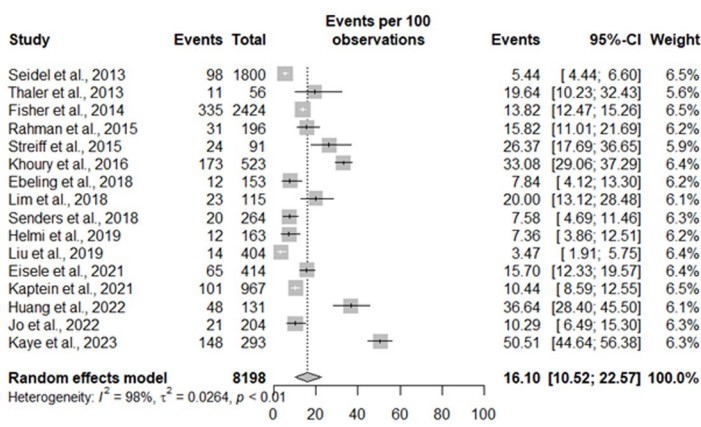

(2C) Forest plot of incidence and 95% confidence interval of venous thromboembolism in a subgroup of patients diagnosed with GBM.

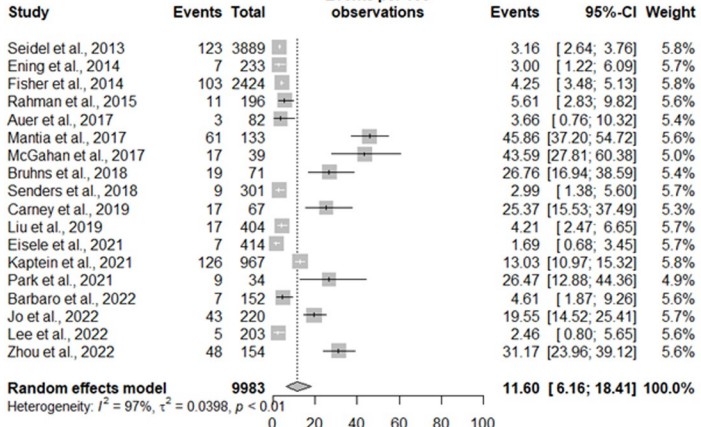

(2B) Forest plot of incidence and 95% confidence interval of bleeding in overall patients with malignant CNS neoplasm.

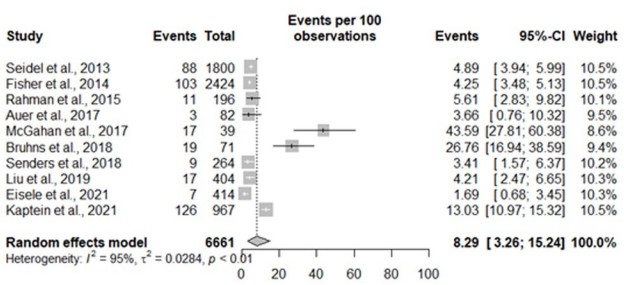

(2D) Forest plot of incidence and 95% confidence interval of bleeding in a subgroup of patients diagnosed with GBM.

**Fig 2. Forest plots of pooled incidence and 95% confidence interval of venous thromboembolism and bleeding in patients with overall malignant primary CNS neoplasm and GBM subgroup.** (2A) Forest plot of incidence and 95% confidence interval of venous thromboembolism in overall patients with malignant CNS neoplasm; (2B) Forest plot of incidence and 95% confidence interval of bleeding in overall patients with malignant CNS neoplasm; (2C) Forest plot of incidence and 95% confidence interval of venous thromboembolism in a subgroup of patients diagnosed with GBM; (2D) Forest plot of incidence and 95% confidence interval of bleeding in a subgroup of patients diagnosed with GBM.

regardless of being benign or malignant, recurrent or metastatic [51–53]. Qian et al. [53] analyzed nine studies and showed that brain tumors, especially those diagnosed with HGG and GBM and submitted to neurosurgery, are associated with an increased risk of VTE. However,

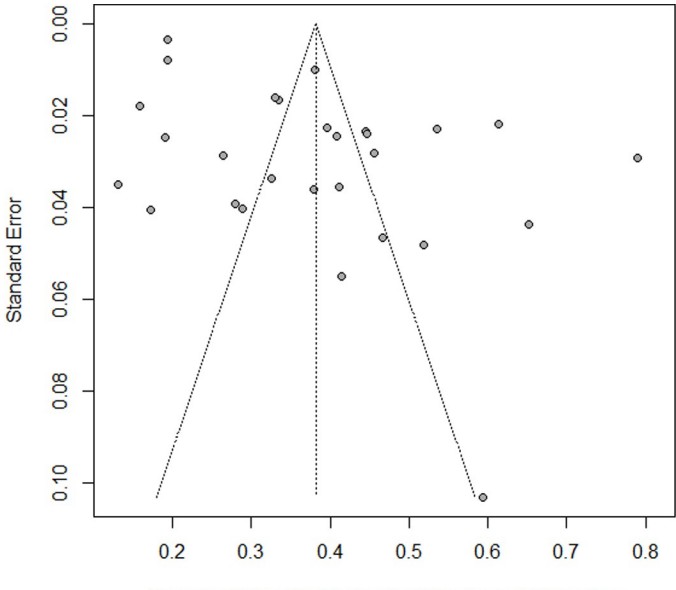

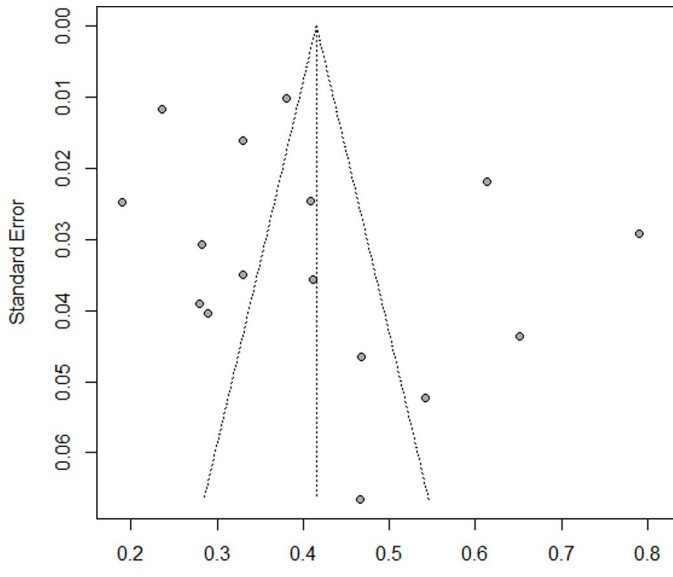

(3A) Funnel plot of Freeman-Tukey Double Arcsine Transformed Proportion using the random effects model for venous thromboembolism in overall patients with CNS malignant neoplasm.

(3C) Funnel plot of Freeman-Tukey Double Arcsine Transformed Proportion using the random effects model for venous thromboembolism in patients with diagnosed GBM.

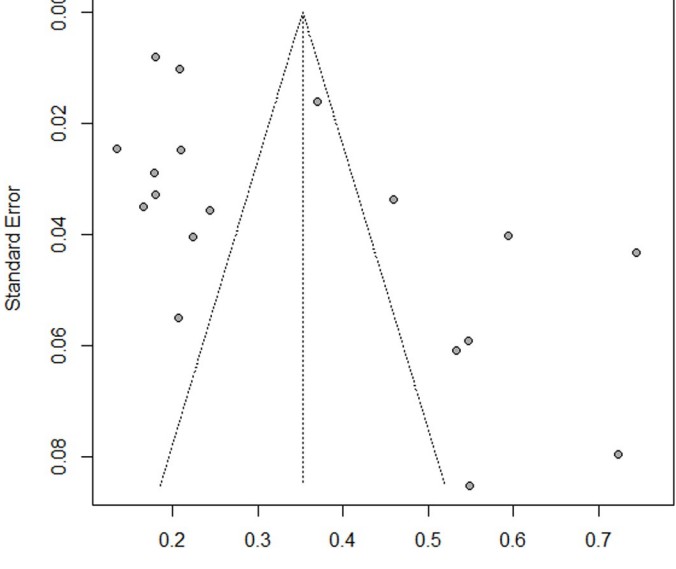

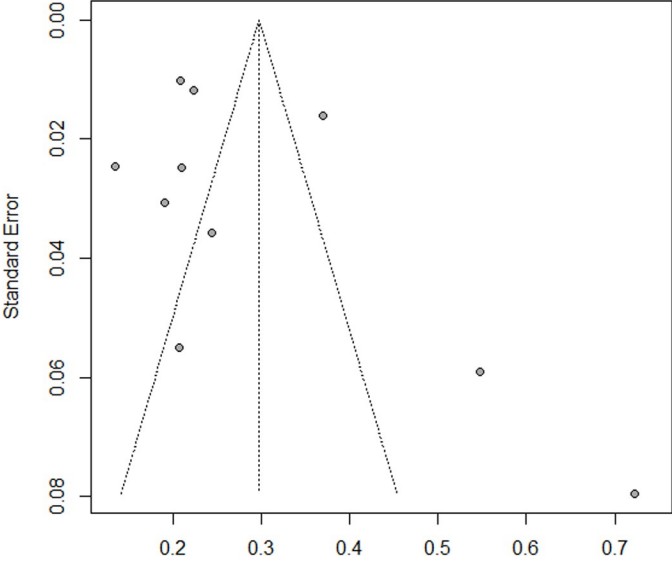

(3B) Funnel plot of Freeman-Tukey Double Arcsine Transformed Proportion using the random effects model for bleeding in overall patients with CNS malignant neoplasm.

(3D) Funnel plot of Freeman-Tukey Double Arcsine Transformed Proportion using the random effects model for bleeding in patients with diagnosed GBM.

**Fig 3. Funnel plots analysis of venous thromboembolism and bleeding in patients with overall malignant primary CNS neoplasm and GBM subgroup.** (3A) Funnel plot of Freeman-Tukey Double Arcsine Transformed Proportion using the random effects model for venous thromboembolism in overall patients with CNS malignant neoplasm; (3B) Funnel plot of Freeman-Tukey Double Arcsine Transformed Proportion using the random effects model for bleeding in overall patients with CNS malignant neoplasm; (3C) Funnel plot of Freeman-Tukey Double Arcsine Transformed Proportion using the random effects model for venous thromboembolism in patients with diagnosed GBM; (3D) Funnel plot of Freeman-Tukey Double Arcsine Transformed Proportion using the random effects model for bleeding in patients with diagnosed GBM.

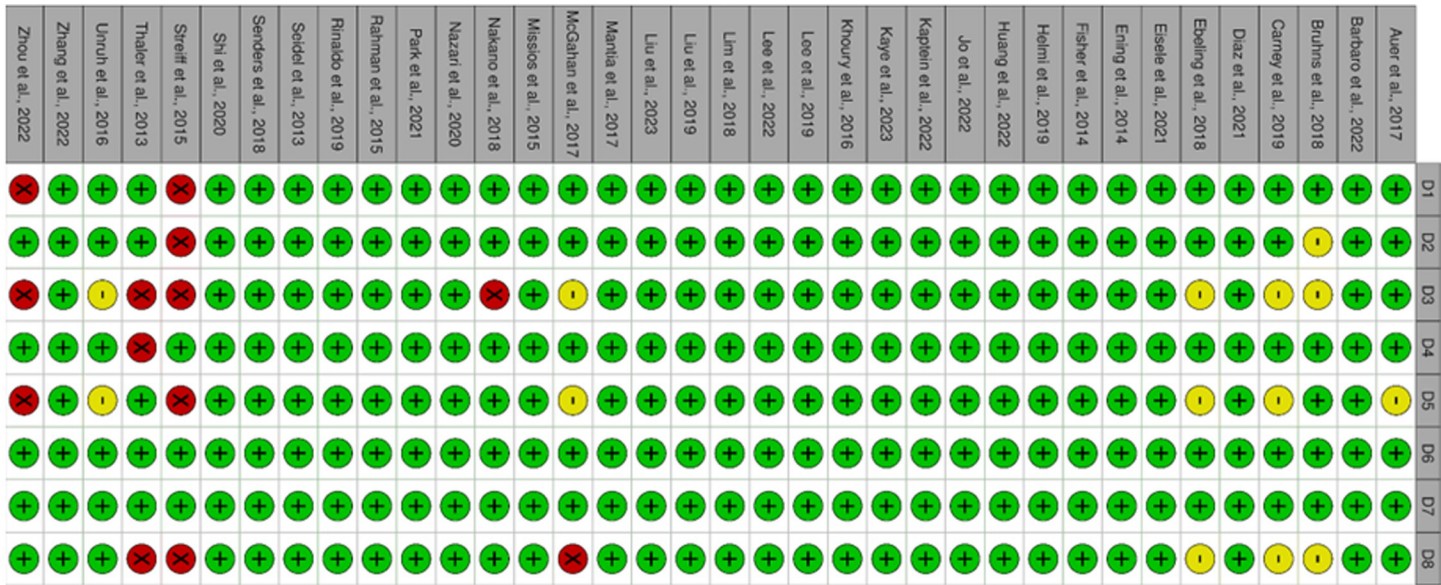

(A) Traffic-light plot of risk of bias assessment per study.

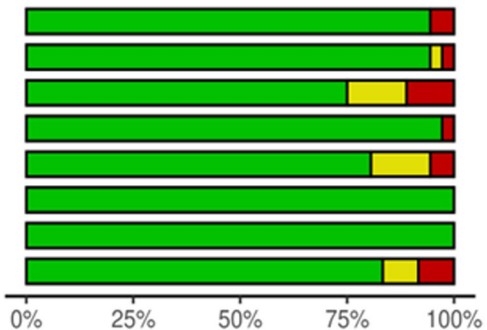

1 Was the sample frame appropriate to address the target population?
2 Were study participants sampled in an appropriate way?
3 Was the sample size adequate?
4 Were the study subjects and the setting described in detail?
5 Was the data analysis conducted with sufficient coverage of the identified sample?
6 Were valid methods used for the identification of the condition?
7 Was the condition measured in a standard, reliable way for all participants?
8 Was there appropriate statistical analysis?

(B) Summary plot of risk of bias assessment. Item 9 (i.e., Was the response rate adequate, and if not, was the low response rate managed appropriately?) was considered not applicable.

**Fig 4. Risk of bias assessment.** (A) Traffic-light plot of risk of bias assessment per study. (B) Summary plot of risk of bias assessment. Item 9 (i.e., Was the response rate adequate, and if not, was the low response rate managed appropriately?) was considered not applicable.

the authors did not mention the prevalence of VTE among these patients [52]. Horsted, West and Grainge [54] pointed out that having brain cancer lead to the second highest risk of VTE inferior only to pancreas cancer. When considering prevalence data, Sun et al. [52] found a prevalence of 7% of VTE in cancer patients undergoing chemotherapy, but only eight among the 102 studies were conducted with brain cancer patients. Regarding brain cancer, the prevalence was ranged between 4 and 5% with low heterogeneity, including five studies with patients diagnosed with recurrent gliomas [52]. Other systematic reviews had effectiveness and safety of prophylaxis and/or treatment for reducing VTE and its complications as purpose

**Table 3. Summary of venous thromboembolism and bleeding incidence according to each step of sensitivity analysis.**

|  | Original | | High quality studies | | Without outliers | |
|---|---|---|---|---|---|---|
|  | Pooled Incidence (95%CI) | $I^2$ | Pooled Incidence (95%CI) | $I^2$ | Pooled Incidence (95%CI) | $I^2$ |
| **CNS** |  |  |  |  |  |  |
| VTE | 13.68 (9.79;18.09) | 99% | 11.72 (8.33;15.59) (S2b Fig in S1 File) | 98% | 18.25 (14,95;21.79) (S2e Fig in S1 File) | 92% |
| Bleeding | 11.60 (6.16;18.41) | 97% | 7.57 (4.08;11.97) (S2c Fig in S1 File) | 95% | 28.42 (23,58;33.51) (S2f Fig in S1 File) | 0% |
| **GBM** |  |  |  |  |  |  |
| VTE | 16.10 (10.52;22.57) | 98% | 13.15 (8.51;18.59) (S2d Fig in S1 File) | 97% | 20.41 (14.79;26.66) (S2g Fig in S1 File) | 95% |
| Bleeding | 8.29 (3.26; 15.24) | 95% | NA | NA | 3.88 (2.96;4.92) (S2h Fig in S1 File) | 50% |

CNS: Central Nervous System; CI: Confidence interval; GBM: Glioblastoma Multiforme; VTE: Venous Thromboembolism.

[48, 55–59]. For bleeding, we observed similar publications that focused on risk of bleeding when submitting patients to a thromboprophylaxis [60]. Accordingly, this is the first systematic review and meta-analysis that provided incidence of both VTE and bleeding in adults with malignant CNS neoplasm.

In the present study, the pooled incidence of VTE was similar to values previously reported [54]. It is worth noting that the comparison of our pooled incidence with those found for Horsted, West and Grainge [54] is limited due to their methodological choice for dividing the studies based on average and high risk. When considering the value found by Sun et al. [52], we must analyze this difference cautiously. Despite their description as brain tumor, most studies were based on recurrent gliomas, especially GBM. Stage of cancer, recurrence, metastasis, and neurosurgery may also play a role in incidence of VTE. This finding may be attributed to the shorter exposure time of these patients to the risk of a VTE. Kaptein et al. [27] showed that the median recurrence among GBM was nearly 8 months, varying between 4.8 months and one year. In newly diagnosed HGG patients, Thaler et al. [61] showed that the probability of having a VTE ranged between 3.3 and almost 40% when considering cancer-related factors (e.g., leukocyte and platelet count, P-selectin, prothrombin-fragment 1 + 2, FVIII activity, and D-dimer).

For GBM, the pooled incidence of VTE was higher than that of overall CNS malignant neoplasms. Due to the hypercoagulability induced by the malignancy, intravascular thrombosis is more frequently observed in GBM cases when compared to other malignant CNS tumors [62]. Conventional treatment stages (e.g., tumor resection, chemotherapy, and radiotherapy) can be considered risk factors for both VTE and bleeding [9, 30, 63]. These factors may contribute to the greater incidence in patients with GBM.

In addition, the risk for VTE is higher in the first two months after surgery in GBM patients [64]. During the first month, the incidence may reach up to 47%, within the period immediately after surgery, it is 40% [64]. Pulmonary embolism was observed in 60% of the cases, with 13% mortality [64]. Patients with GBM are included in the three groups with the highest risk of thromboembolic complications, in addition to those with pancreas, liver and ovarian cancer [65, 66].

Although the incidence of malignant CNS tumors is higher in men [67], the incidence of VTE and bleeding was similar when considering the subgroup analysis, corroborating previous investigations [31, 32, 68]. Mulder et al. [69] assessed the occurrence of VTE in the first six months of the follow-up and found no difference between the sexes in sub-distribution hazard ratios (SHR = 1.02; 0.98–1.07). The cumulative incidence was 1.61 (95%CI 1.56–1.66) in

women and 1.78 (95%CI 1.73–1.83) in men. This may be attributed to the severity of the disease, which exposes both sexes to similar risk factors.

In this systematic review and meta-analysis, the highest incidence of VTE and/or bleeding was observed in older adults ($\geq$ 60 years). Aging may increase the risk of VTE. In patients with cancer, the occurrence of VTE can increase by up to 3 times when compared to their younger counterparts [70, 71]. In a population-based case-control study of older adults with different types of cancer, the likelihood of thrombotic events increased between 27 and 92% [72]. In a cohort study involving patients with and without cancer, who suffered from VTE, fatal bleeding occurred in 0.8% of the older adults and 0.4% of their younger counterparts, resulting in an HR equal to 2.0 (95% CI = 1.2–3.4) [73]. Regardless of cancer type, aging is associated with an increased risk of thrombosis, particularly due to reduced physical activity, a decline in mobility, greater disability in activities of daily living and systemic activation of coagulation.

Asian and European countries present a greater incidence and health disorders caused by CNS tumors [4], which may explain the larger number of studies from these regions. The age-related incidence is greater in North American and European countries [4], which also agree with our findings. Incidence according to socioeconomic subgroup based on World Banking classification differed between high-income and upper-middle income countries. Despite being considered a health problem, especially in high-income countries, the lack of an opportune and accurate diagnosis of CNS neoplasm in low-income regions may explain the lower incidence rates and, consequently, result in less access to treatment and higher disability and mortality [4]. Thus, the absence of studies from low and lower-middle income countries could also justify these findings. Moreover, cancer-related risk factors for VTE include brain cancer (as a high risk for developing VTE), as well as stage of cancer, especially advanced stage, and active treatment, while being older with black ethnicity and presence of comorbidities are patient-related risk factors [74]. Briefly, these patient-related factors are often found among low and lower-middle income households and linked with over 80% of premature deaths [75], which can be even more alarming when considering the lack of data for these populations. However, the healthcare access is limited and commonly associated with ageing in a context of poverty and income inequality [76].

This systematic review and meta-analysis presented limitations and strengths. Despite the studies included, the lack of detailed sample descriptions hampered subgroup analyses for both sociodemographic variables and those related to CNS neoplasm. Moreover, although the studies were cohorts, the retrospective design predominated, and cohort and case-control nested studies were included, which may be due to the less detailed sample descriptions and treatments. It is worth nothing that the studies often presented the events according to the total sample, which, at times, consisted of more than one age range, both sexes and different types of CNS tumors.

Among the strengths are the comprehensive search strategy and the adoption of highly inclusive eligibility criteria, which provided a general overview of the literature. We also carried out a sensitivity analysis and apply an instrument to assess methodological quality and risk of bias as recommended. Sensitivity analyses followed two distinct patterns. Outlier exclusion was guided by a methodological rationale, specifically targeting studies where the event incidence was 1.5 times greater than the interquartile range. This analysis demonstrated a pooled incidence rate of VTE in GBM tumors consistent with that reported in the literature (20.41%) [9]. The second approach involved subgroup analysis, categorizing groups based on demographic variables (such as sex and age) and tumor type. In the studies reviewed, older age emerged as the primary variable identified as a risk factor for both VTE outcomes and bleeding. Performance status was an independent risk factor in three articles, albeit with variations

in instruments and data collection methodologies. Nevertheless, it was noted that several studies lacked multivariate analyses, either due to the lack of significance in univariate analysis, the size of sample subgroups, or the specific objective of investigating a particular tumor marker.

The substantial heterogeneity observed should be analyzed with caution. A high $I^2$ is expected due to the study design (i.e., meta-analysis of incidence) and does not necessarily imply either relevant heterogeneity or the absence of a conclusion [77]. The interstudy variability may be attributed to the lack of standardized data collection and registration, types of malignant tumors and diagnostic methods, especially in asymptomatic cases, cancer stage, type of treatment, the introduction of thromboprophylaxis and other patient-related factors. In addition, it is important to consider different concepts for defining bleeding and its prognosis [77], particularly in postoperative cases. For instance, we were unable to analyze whether most cases actually involved major bleeding (i.e., clinically overt bleeding), clinically relevant non-major bleeding (i.e., episode associated with medical intervention that did not meet the criteria for major bleeding, which can affect treatment continuation and compromise patients' activities of daily living) or minor bleeding potentially misclassified since the amount of bleeding and site can influence the clinical outcome (serious/disabling, severe and life-threatening) [78].

The findings suggest gaps in the literature regarding the influence of tumor type and characteristics on the incidence of events of interest, especially investigating possible confounders and biases, for instance, the characteristics inherent to surgical procedures (type of procedure, duration, presence and volume of bleeding, pre, peri- and/or post-operatory, prophylactic measures, possible complications, among others), active cancer treatment (radiotherapy and separate or concomitant chemotherapy) and health history (including mapping comorbidities). Future perspectives indicate the need for scientific knowledge on the topic in low-income countries with greater social inequality, making it possible to obtain incidence data in these regions, thereby favoring greater understanding of the role of the social determinants of health.

## Conclusion

According to this research, the pooled incidence showed variability across all analyses and their subgroups for both events. Subgroup analysis showed that being older than 60 years or having GBM diagnosis presented higher pooled incidence values in comparison to overall CNS malignant neoplasm. In addition to the sensitivity analysis, when considering the outlier criterion, it's noted a higher pooled incidence among GBM, mirroring findings in the literature. Further studies from low and lower-middle income countries should be encouraged.

## Supporting information

**S1 Appendix. REDCap file for data extraction.**
(DOCX)

**S2 Appendix. Guidance for screening and data extraction.**
(DOCX)

**S1 Checklist.**
(DOCX)

**S1 File. Forest and funnel plots of venous thromboembolism and bleeding according sensitivity analysis, and 95%CI graph to sensitivity analysis groups.**
(ZIP)

**S2 File.** a-f. Forest plots of venous thromboembolism and bleeding according subgroup analysis.
(ZIP)

**S1 Table. Search strategy according to electronic databases.**
(DOCX)

**S2 Table. Characteristics of excluded studies (*ordered by study ID*).**
(DOCX)

**S3 Table. Conflict of interest and funding reported in the included studies.**
(DOCX)

## Acknowledgments

The authors are grateful to Brazilian Ministry of Health for their support. We also would like to thank all the other researchers who are part of the TROMBOGLIO Study Group (i.e., Ana Carolina Sigolo Levy, Silvana Soares dos Santos, Sabrina Dos Santos Pinho Costa, Jessica Carolina Andrade dos Santos, Carolina Fittipaldi Pessoa, Daniela Galvão Barros de Oliveira, Thiago Santos Vieira, Danielli de Almeida Matias, Kleyton Medeiros, Alessandra Buccaran, Breno Gray Milano, Raiane Alves da Costa, and Fabiana Spillari Viola).

## Author Contributions

**Conceptualization:** Viviane Cordeiro Veiga, Flavia Regina Moraes, Talita Rantin Belucci, Camilla Akemi Felizardo Yamada.

**Data curation:** Viviane Cordeiro Veiga, Stela Verzinhasse Peres, Thatiane L. V. D. P. Ostolin, Flavia Regina Moraes, Camilla Akemi Felizardo Yamada.

**Formal analysis:** Stela Verzinhasse Peres, Thatiane L. V. D. P. Ostolin.

**Funding acquisition:** Viviane Cordeiro Veiga, Flavia Regina Moraes.

**Investigation:** Stela Verzinhasse Peres, Thatiane L. V. D. P. Ostolin.

**Methodology:** Stela Verzinhasse Peres, Thatiane L. V. D. P. Ostolin.

**Project administration:** Flavia Regina Moraes.

**Resources:** Viviane Cordeiro Veiga, Flavia Regina Moraes, Camilla Akemi Felizardo Yamada.

**Software:** Stela Verzinhasse Peres, Thatiane L. V. D. P. Ostolin.

**Supervision:** Viviane Cordeiro Veiga, Flavia Regina Moraes.

**Validation:** Viviane Cordeiro Veiga, Flavia Regina Moraes, Camilla Akemi Felizardo Yamada.

**Visualization:** Viviane Cordeiro Veiga.

**Writing – original draft:** Stela Verzinhasse Peres, Thatiane L. V. D. P. Ostolin.

**Writing – review & editing:** Viviane Cordeiro Veiga, Stela Verzinhasse Peres, Thatiane L. V. D. P. Ostolin, Flavia Regina Moraes, Talita Rantin Belucci, Carlos Afonso Clara, Alexandre Biasi Cavalcanti, Feres Eduardo Aparecido Chaddad-Neto, Gabriel N. de Rezende Batistella, Iuri Santana Neville, Alex M. Baeta, Camilla Akemi Felizardo Yamada.

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
