## [Editor Report · Decision Letter 0]

11 Jan 2024

PONE-D-23-39566Incidence of venous thromboembolism and bleeding in patients with malignant central nervous system neoplasm: systematic review and meta-analysisPLOS ONE

Dear Dr. Peres,

Thank you for submitting your manuscript to PLOS ONE. As indicated in my previous E-mail to you, we need you to submit the manuscript with high-quality figures. Note that the manuscript did not go through the full review process as it was interrupted based on a reviewer comment about the figures quality which hindered the completion of his/her review. Please, submit the manuscript with high-quality figures so we can continue the review process.

Kind regards,

Omar A. Almohammed, Ph.D.

Academic Editor

PLOS ONE

3. Please include your tables as part of your main manuscript and remove the individual files. Please note that supplementary tables (should remain/ be uploaded) as separate ""supporting information"" files.

---

## [Author Response · Author response to Decision Letter 0]

15 Jan 2024

Dear Editor,

We have corrected the file by inserting the high-quality figure into the submission system. For a better resolution, we suggest to download the figures before revising.

Best regards,

Stela Verzinhasse Peres

---

## [Decision Letter · Decision Letter 1]

7 Feb 2024

PONE-D-23-39566R1Incidence of venous thromboembolism and bleeding in patients with malignant central nervous system neoplasm: systematic review and meta-analysisPLOS ONE

Dear Dr. Peres,

Thank you for submitting your manuscript to PLOS ONE. After careful consideration, we feel that it has merit but does not fully meet PLOS ONE’s publication criteria as it currently stands. Therefore, we invite you to submit a revised version of the manuscript that addresses the points raised during the review process. Please submit your revised manuscript by Mar 23 2024 11:59PM. If you will need more time than this to complete your revisions, please reply to this message or contact the journal office at plosone@plos.org. Please include the following items when submitting your revised manuscript:A rebuttal letter that responds to each point raised by the academic editor and reviewer(s). You should upload this letter as a separate file labeled 'Response to Reviewers'.A marked-up copy of your manuscript that highlights changes made to the original version. You should upload this as a separate file labeled 'Revised Manuscript with Track Changes'.An unmarked version of your revised paper without tracked changes. You should upload this as a separate file labeled 'Manuscript'.

We look forward to receiving your revised manuscript.

Kind regards,

Omar A. Almohammed, Ph.D.

Academic Editor

PLOS ONE

Additional Editor Comments:

You will see that one of the reviewer was not able to see that high quality figures you submitted. However, I decided to send it back to you anyway to save you some time and address the other comments and leave that for another round of review.

Reviewers' comments:

Reviewer's Responses to Questions

**Comments to the Author**

1. If the authors have adequately addressed your comments raised in a previous round of review and you feel that this manuscript is now acceptable for publication, you may indicate that here to bypass the “Comments to the Author” section, enter your conflict of interest statement in the “Confidential to Editor” section, and submit your "Accept" recommendation.

Reviewer #1: (No Response)

Reviewer #2: (No Response)

2. Is the manuscript technically sound, and do the data support the conclusions?

Reviewer #1: Partly

Reviewer #2: Partly

3. Has the statistical analysis been performed appropriately and rigorously? 

Reviewer #1: No

Reviewer #2: Yes

4. Have the authors made all data underlying the findings in their manuscript fully available?

Reviewer #1: No

Reviewer #2: Yes

5. Is the manuscript presented in an intelligible fashion and written in standard English?

Reviewer #1: Yes

Reviewer #2: No

6. Review Comments to the Author

Reviewer #1: Abstract:

• (Line 48)- CI 95%; needs edit

• (Line 53-54)- Vague sentence

Introduction:

Well written.

• (Line 81-83): Vague sentence

Methods:

(Line 93)- Cite your source and mention what CoCoPop stands for.

(Line-97) - Needs edit

(Line -106) - What do you mean by primary cohort?

(Line-107) - While your study's aim was to pool incidence, why did you include a case-control study from which we cannot calculate incidence?

(Line-108) – What do you mean by “Secondary studies”?

(Line 113-114) – Explain why you excluded those types of studies?

(Line -123) – Specialists in what area? Specify it.

(Line -125) –Specifically describe those “minimum adjustments” the search strategy underwent

Results:

(Line-196) - Mention only the most recent time you conducted the search, or explain why you conducted the search at two different times.

(Line-198) – Specify which “software”

(Line-202) – There are only 28 studies displayed in the overall forest plot (Figure 2, Forest plot-A), not 36. Why is this so?

(Line-203)- “15-49” references result in 35 studies, not 36.

(Line-260)- Cite the “34-persons” study and the “21384-persons” study. Nakano et al., 2018 included 23 research participants rather, which is fewer than the 34 you list as the smallest study; please explain it.

(Line-261-262)- Indicate the percentage and mention the particular age that pertains to "older"

(Line 288-296)- The forest plots A-D are not identified properly (there is no heading).

(Line 298-299) - Sub-group analyses report should be supported by their respective forest plots.

(Line – 361) – Unlike risk ratio or odds ratio, incidence is not a measure of impact/association that provides the direction of effect. Correct your statement.

(Line - 362)- Include a graph representing the influential study analysis.

Discussion:

Well discussed, but it needs some edits

Kind regards,

Reviewer #2: General comments

- There are many (more than 2) minor grammatical errors throughout the manuscript. Some of the examples are

1. 'The used the following' definition for VTE

2. Then, the files were saved in Rayyan® software 'to screening and eligibility.'

Introduction

- In general, the introduction is not convincing. These suggestion may be helpful.

- Please provide information of other common types of CNS tumors. Please also explain whether VTE and bleeding are also found in these common CNS tumors. This is to ensure that the results from different types of tumors should be homogenous or heterogenous.

- The introduction states that the risk of bleeding is from drugs. Please provide brief background information for the prophylaxis used in neurosurgery and their risk of bleeding. Please also provide information on the risk of bleeding from the tumors. This is to clearly mention possible confounders.

Methods

- The protocol for this systematic review, which was submitted to PROSPERO, stated that Embase, BVS, OpenGrey, medRxiv and Google Scholar would also be used. However, these databases are not mentioned in the Methods. Please correct this section and the PRISMA diagram.

- The study search was limited to the period between 2013 and 2023. --- Please provide the rationale for this time frame.

- According to the Discussion, the authors mention that literature has found that type of CNS tumor (especially GBM), duration after surgery, and sex affect the incidence of VTE. This information should be in the Introduction. Also, subgroup analysis or meta-regression based on these factors should be conducted. If they are not possible, please explain why subgroup analysis or meta-regression are not possible and discuss the potential confounding effects of these factors in the discussion.

Results

- Table 1 should provides information on confounders and information used for assessing risk of bias.

- The quality of figure 2-4 are low to the unreadable point. Please improve the quality.

Discussion

- Please reduce the length of this part and focus more on the potential confounders that affect the incidence of VTE and bleeding.

7. PLOS authors have the option to publish the peer review history of their article (what does this mean?). If published, this will include your full peer review and any attached files.

Reviewer #1: No

Reviewer #2: No

---

## [Author Response · Author response to Decision Letter 1]

21 Mar 2024

Dear Editor,

The responses to the reviewers' comments are found in the file "Letter_revise_editor".

Revisor 1:

Dear Reviewer,

Thank you for your contribution to improve this manuscript, and we agree with your comments. All modifications in the manuscript have been highlighted in red in the text.

Abstract:

• (Line 48)- CI 95%; needs edit

• (Line 53-54)- Vague sentence

Answer: Thanks for the consideration, we adjusted this point.

Line 48 – page 3

Regarding vague sentence, we aim was to evidence the absent studies in lower and middle-income. Thus, we changed the conclusion for:

Line 53-56 – page 3

“It is important to note that the results of this meta-analysis refer mainly to studies carried out in high-income countries. This highlights the need for additional research in Latin America, and low- and middle-income countries.”

Introduction:

Well written.

• (Line 81-83): Vague sentence

Answer: 

Thank you for bringing this to our attention. We improved this paragraph for clarified and we included more information.

Line 80-95 – page 4 and 5.

“Among CNS tumors, GBM with wild type isocitrate dehydrogenase (IDH) has a poorer prognosis and higher incidence of thrombotic events, estimated at approximately 20-30% per year (9). 

The increased risk of venous thromboembolism (VTE) in cancer patients is particularly notable, and in neurosurgery, the introduction of pharmacological prophylaxis is well established in the literature and should be instituted 24 hours after the procedure (10,11). Given the substantial pathogenic propensity inherent to central nervous system (CNS) neoplasms, triggering thrombotic events such as ischemic stroke, myocardial infarction, peripheral arterial disease, and deep vein thrombosis (DVT), anticoagulants are a preventive measure against these potential risks. Conversely, the occurrence of minor or major hemorrhagic events in internal organs is closely connected to the underlying pathological condition of the disease and may be exacerbated by the prophylactic or therapeutic administration of anticoagulants. In this clinical milieu, it is imperative to meticulously assess and balance the inherent risk associated with these two complications.”

Methods:

(Line 93)- Cite your source and mention what CoCoPop stands for.

Answer: Thank you for consideration. We adjusted in: 

Line 105-106 – page 5

“… Condition, Context and Population (CoCoPop), i.e. …” 

(Line-97) - Needs edit

Answer: We changed the paragraph for:

Line 110-113 – page 6

“… VTE: any symptomatic or incidental event involving the upper or lower limbs, confirmed by imaging examinations such as venous Doppler ultrasound and/or computerized tomography of the lungs, lung scintigraphy, and angiography.”

(Line -106) - What do you mean by primary cohort?

Answer: Dear reviewer, it was a translation error that proposed the joining of two information from cohort studies of primary CNS neoplasms.

All the paragraph was corrected.

Line 118-127 – page 6

“The following were considered eligible: (1) cohort studies (prospective and retrospective), case-control nested cohort studies, and cohort nested case-control studies that (2) assessed the presence of VTE and bleeding in (3) patients with malignant primary CNS neoplasm. The studies were deemed eligible when presenting, at least, a numerator and denominator for the total sample to calculate the event of interest. Review and metanalysis studies, letters to the editor, opinion articles, comments, short communications, ecological studies, and abstracts published in the annals of scientific events were not included. Studies that investigated metastatic tumors, CNS lymphoma and meningioma below grade 3 were also excluded.”

(Line-107) - While your study's aim was to pool incidence, why did you include a case-control study from which we cannot calculate incidence?

Answer: Thank you for the careful evaluation that provided to clarify the points of this manuscript.

We agreed with your position. However, these studies presented numerator and denominator, and it was nested studies.

Line 119-120 – page 6

“… case-control nested cohort studies, and cohort nested case-control studies that …”

(Line-108) – What do you mean by “Secondary studies”?

Answer: We classified we inappropriately classified meta-analysis and review studies that were excluded.

Line 123 – page 6

“… Review and metanalysis studies …”

(Line 113-114) – Explain why you excluded those types of studies?

Answer: Thank you for consideration. We will elucidate this methodological consideration due to clinical issues.

The objective was to minimize heterogeneity concerning tumor type and malignancy. Another aspect to consider is time. This manuscript was developed to support research with the Brazilian Ministry of Health; thus, its execution was confined to the specified timeframes within the study.

(Line -123) – Specialists in what area? Specify it.

Answer: We detailed this information in:

Line 138-139 – page 7

“… , physicians with a specialization in intensive care and neuro-oncologists, and oncology nurses …” .

Line 140-141 – page 7

“ …(VCV, CAFY, and FRdM, respectively)….”

(Line -125) –Specifically describe those “minimum adjustments” the search strategy underwent

Answer: Thank you for consideration, and we insert the paragraph the minimum adjustments

Line 142-150 – page 7

“… Although standardized, the search strategy underwent minimum adjustments (i.e., minor differences in filters, for instance, using Full text, Case Reports, and Observational Studies, in the last 10 years, in English, Portuguese, Spanish, in Adults 19 years or older. Excluded were preprints as filters for PubMed, while fulltext, type_of_study:"observational_studies" OR "incidence_studies" OR "prevalence_studies", la:"en" OR "es" OR "pt", and year cluster for BVS) according to the database investigated as shown in the supplementary file (S1 Table. Search strategy according to electronic databases). However, the search focused on MeSH terms for all databases since PubMed, BVS and Cochrane considered them controlled vocabulary…”

Results:

(Line-196) - Mention only the most recent time you conducted the search, or explain why you conducted the search at two different times.

Answer: Thank for suggestion. The study was conducted over a single period. We rephrased the sentence in the text for better understanding.

Line 226 – page 10

“Database searches were carried from June to July 2023.”

(Line-198) – Specify which “software”

Answer: Thank for observation.

Line 228 – page 10

“… were identified by the Rayyan® software… “

(Line-202) – There are only 28 studies displayed in the overall forest plot (Figure 2, Forest plot-A), not 36. Why is this so?

Answer: Thank you for observation, and this forest plot is relationship to VTE. For clarified, we put on this information in the text.

Line 233 – page 10

“… 28 showed information for VTE and 18 for bleeding. …”

(Line-203)- “15-49” references result in 35 studies, not 36.

Answer: Thank you for observation, we complete with reference at Zhang et al.

Line 233 – page 10

“ 50. Zhang C, Deng Z, Yang Z, Xie J, Hou Z. A nomogram model to predict the acute venous thromboembolism risk after surgery in patients with glioma. Thromb Res. 2023 Apr;224:21-31. doi: 10.1016/j.thromres.2023.02.002. Epub 2023 Feb 9. PMID: 36805800.”

(Line-260)- Cite the “34-persons” study and the “21384-persons” study. Nakano et al., 2018 included 23 research participants rather, which is fewer than the 34 you list as the smallest study; please explain it.

Answer: We corrected the information to 23 patients.

Line 296 – page 19

(Line-261-262)- Indicate the percentage and mention the particular age that pertains to "older"

Answer: Thank you for consideration, we included a paragraph.

Line 298-300 – page 19

“… Only three studies presented data on patients aged 60 years or older (28,31-32), totaling 163 patients with VTE out of 446 patients with CNS (36.5%). …”

(Line 288-296)- The forest plots A-D are not identified properly (there is no heading).

Answer: Thank you for consideration. We adjusted the picture. 

Line 327 – page 20

(Line 298-299) - Sub-group analyses report should be supported by their respective forest plots.

Answer: Thank you for consideration. The forest-plots were attached in Figure 2. 

Line 324-326 – page 20

“… others subgroup analyzes were summarized in Table 2 (Figure 2). The forest plots for subgroup analysis can be found in the supplementary material (S1 Figures a-f).”

(Line – 361) – Unlike risk ratio or odds ratio, incidence is not a measure of impact/association that provides the direction of effect. Correct your statement.

Answer: Thank you for the careful evaluation. We agree and emphasize that the study does not aim to compare subgroups.

The information was excluded from the text.

(Line - 362)- Include a graph representing the influential study analysis.

Answer: We created a graph of the quality of the studies that can be found in the Figure 4 material, and we include in S2 Forest and funnel plots of venous thromboembolism and bleeding according sensitivity analysis, and 95%CI graphic to sensitivity analysis groups).

Reviewer #2--------------------------------------------------

General comments

- There are many (more than 2) minor grammatical errors throughout the manuscript. Some of the examples are

1. 'The used the following' definition for VTE

2. Then, the files were saved in Rayyan® software 'to screening and eligibility.'

Answer: Thank you for consideration. This manuscript was resubmitted for textual review.

Introduction

- In general, the introduction is not convincing. These suggestion may be helpful.

- Please provide information of other common types of CNS tumors. Please also explain whether VTE and bleeding are also found in these common CNS tumors. This is to ensure that the results from different types of tumors should be homogenous or heterogenous.

- The introduction states that the risk of bleeding is from drugs. Please provide brief background information for the prophylaxis used in neurosurgery and their risk of bleeding. Please also provide information on the risk of bleeding from the tumors. This is to clearly mention possible confounders.

Answer: Thank you for pointing this out. We improved this paragraph for clarified and we included more information.

Line 80-95 – page 4 and 5.

“Among CNS tumors, GBM with wild type isocitrate dehydrogenase (IDH) has a poorer prognosis and higher incidence of thrombotic events, estimated at approximately 20-30% per year (9). 

The increased risk of venous thromboembolism (VTE) in cancer patients is particularly notable, and in neurosurgery, the introduction of pharmacological prophylaxis is well established in the literature and should be instituted 24 hours after the procedure (10,11). Given the substantial pathogenic propensity inherent to central nervous system (CNS) neoplasms, triggering thrombotic events such as ischemic stroke, myocardial infarction, peripheral arterial disease, and deep vein thrombosis (DVT), anticoagulants are a preventive measure against these potential risks. Conversely, the occurrence of minor or major hemorrhagic events in internal organs is closely connected to the underlying pathological condition of the disease and may be exacerbated by the prophylactic or therapeutic administration of anticoagulants. In this clinical milieu, it is imperative to meticulously assess and balance the inherent risk associated with these two complications.”

Methods

- The protocol for this systematic review, which was submitted to PROSPERO, stated that Embase, BVS, OpenGrey, medRxiv and Google Scholar would also be used. However, these databases are not mentioned in the Methods. Please correct this section and the PRISMA diagram.

Answer: We appreciate the suggestion, but we chose to create a chapter on protocol deviations to clarify the reasons for the changes proposed in PROSPERO.

Line 203-224 – pages 9 and 10.

Line 80-95 – page 4 and 5.

“Among CNS tumors, GBM with wild type isocitrate dehydrogenase (IDH) has a poorer prognosis and higher incidence of thrombotic events, estimated at approximately 20-30% per year (9). 

The increased risk of venous thromboembolism (VTE) in cancer patients is particularly notable, and in neurosurgery, the introduction of pharmacological prophylaxis is well established in the literature and should be instituted 24 hours after the procedure (10,11). Given the substantial pathogenic propensity inherent to central nervous system (CNS) neoplasms, triggering thrombotic events such as ischemic stroke, myocardial infarction, peripheral arterial disease, and deep vein thrombosis (DVT), anticoagulants are a preventive measure against these potential risks. Conversely, the occurrence of minor or major hemorrhagic events in internal organs is closely connected to the underlying pathological condition of the disease and may be exacerbated by the prophylactic or therapeutic administration of anticoagulants. In this clinical milieu, it is imperative to meticulously assess and balance the inherent risk associated with these two complications.”

- The study search was limited to the period between 2013 and 2023. --- Please provide the rationale for this time frame.

Answer: Thank you for observation. We include the requested information after the description of the period. 

Line 128 -130 – page 6

“… The period was limited to the largest number of publications on the topic and with more homogeneous patient selection criteria.…”

Furthermore, this manuscript was developed to support research with the Brazilian Ministry of Health; thus, its execution was confined to the specified timeframes within the study.

- According to the Discussion, the authors mention that literature has found that type of CNS tumor (especially GBM), duration after surgery, and sex affect the incidence of VTE. This information should be in the Introduction. Also, subgroup analysis or meta-regression based on these factors should be conducted. If they are not possible, please explain why subgroup analysis or meta-regression are not possible and discuss the potential confounding effects of these factors in the discussion.

Answer: Dear reviewer, we included this information in the introduction. We highlight wild-type GBM as well as risk factors. 

(Line 80-95 – page 4 and 5)

Regarding the analysis subgroups, we opted for two types, which we present in the discussion. As suggested, we included factors related to the outcomes of interest in Table 1. When available, we included results of multiple analyses.

Table 1 – page 290

Line 280-288 – page 12 

“Among the studies examined in Table 1, noteworthy findings pertain to the risk factors identified in both univariate and multivariate analyses. Across these studies, advanced age was correlated with the outcomes under investigation in this meta-analysis. Performance status was similarly highlighted in three studies, while a history of VTE was documented in two. Additional studies were specifically chosen to illustrate outcome incidence, although the primary focus on risk factors, along with their corresponding adjustment variables, was directed toward the outcomes of death or recurrence.” 

Line 516-529 – page 28

“Sensitivity analyses followed two distinct patterns. Outlier exclusion was guided by a methodological rationale, specifically targeting studies where the event incidence was 1.5 times greater than the interquartile range. This analysis demonstrated a pooled incidence rate of VTE in GBM tumors consistent with that reported in the literature (20.41%) (9). The second approach involved subgroup analysis, categorizing groups based on demographic variables (such as sex and age) and tumor type. In the studies reviewed, older age emerged as the primary variable identified as a risk factor for both VTE outcomes and bleeding. Performance status was an independent risk factor in three articles, albeit with variations in instruments and data collection methodologies. Nevertheless, it was noted that several studies lacked multivariate analyses, either due to the lack of significance in univariate analysis, the size of sample subgroups, or the specific o

---

## [Decision Letter · Decision Letter 2]

30 Apr 2024

PONE-D-23-39566R2Incidence of venous thromboembolism and bleeding in patients with malignant central nervous system neoplasm: systematic review and meta-analysisPLOS ONE

Dear Dr. Peres,

Thank you for submitting your manuscript to PLOS ONE. After careful consideration, we feel that it has merit but does not fully meet PLOS ONE’s publication criteria as it currently stands. Therefore, we invite you to submit a revised version of the manuscript that addresses the points raised during the review process.

We look forward to receiving your revised manuscript.

Kind regards,

Omar A. Almohammed, Ph.D.

Academic Editor

PLOS ONE

Journal Requirements:

Reviewers' comments:

Reviewer's Responses to Questions

**Comments to the Author**

1. If the authors have adequately addressed your comments raised in a previous round of review and you feel that this manuscript is now acceptable for publication, you may indicate that here to bypass the “Comments to the Author” section, enter your conflict of interest statement in the “Confidential to Editor” section, and submit your "Accept" recommendation.

Reviewer #1: All comments have been addressed

Reviewer #2: All comments have been addressed

2. Is the manuscript technically sound, and do the data support the conclusions?

Reviewer #1: Yes

Reviewer #2: Yes

3. Has the statistical analysis been performed appropriately and rigorously? 

Reviewer #1: Yes

Reviewer #2: Yes

4. Have the authors made all data underlying the findings in their manuscript fully available?

Reviewer #1: Yes

Reviewer #2: (No Response)

5. Is the manuscript presented in an intelligible fashion and written in standard English?

Reviewer #1: Yes

Reviewer #2: Yes

6. Review Comments to the Author

Reviewer #1:

Dear Authors,

Firstly, congratulations on your remarkable work! However, upon review, it appears that some language edits are necessary throughout the document. Additionally, we noticed that your conclusions seem to mirror your results, which might benefit from further refinement to avoid redundancy.

Reviewer #2:

I have no concern regarding this revision. The authors responded to all my comment in a professional manner.

7. PLOS authors have the option to publish the peer review history of their article (what does this mean?). If published, this will include your full peer review and any attached files.

Reviewer #1: **Yes**

Reviewer #2: No

---

## [Author Response · Author response to Decision Letter 2]

14 May 2024

Dear Reviewer,

Thank you for your precious time in reviewing our paper and providing valuable comments.

Reviewer #1:

Dear Authors,

Firstly, congratulations on your remarkable work! However, upon review, it appears that some language edits are necessary throughout the document. Additionally, we noticed that your conclusions seem to mirror your results, which might benefit from further refinement to avoid redundancy.

Answer: Thank you for your consideration and we have changed the paragraphs in abstract, discussion, and conclusion:

Abstract – page 3, lines 51 – 54.

Before “The pooled incidence was 13.68% and 11.60% for VTE and bleeding, respectively. Being older than 60 years or having GBM diagnosis led to pooled incidence of 32.27% and 16.10% for VTE.”

After “Our study showed variability in the pooled incidence for both overall events and subgroup analyses. It was highlighted that individuals over 60 years old or diagnosed with GBM had a higher pooled incidence of VTE among those with overall CNS malignancies.”

Discussion – page 24, lines 436 – 437.

Before “In the present study, the pooled incidence of VTE was 13.68%. The incidence was similar to values previously reported (54)”.

After “In the present study, the pooled incidence of VTE was similar to values previously reported (54).

Discussion – page 25, lines 452 – 452.

Before “For GBM, the pooled incidence was 16.10% (95%CI 10.52; 22.57) for VTE.”

After “For GBM, the pooled incidence of VTE was higher than that of overall CNS malignant neoplasms.”

Conclusion – page 29, lines 559 – 565.

Before “The pooled incidence was 13.68% for VTE. Similarly, the incidence pooled of bleeding was 11.60%. Subgroup analysis showed that being older than 60 years or having GBM diagnosis presented higher pooled incidence values in comparison to overall CNS malignant neoplasm. Further studies from low and lower-middle income countries should be encouraged.”

After “According to this research, the pooled incidence showed variability across all analyses and their subgroups for both events. Subgroup analysis showed that being older than 60 years or having GBM diagnosis presented higher pooled incidence values in comparison to overall CNS malignant neoplasm. In addition to the sensitivity analysis, when considering the outlier criterion, it's noted a higher pooled incidence among GBM, mirroring findings in the literature. Further studies from low and lower-middle income countries should be encouraged.”

Thanks for your consideration.

Sincerely

Stela Verzinhasse Peres

---

## [Editor Report · Decision Letter 3]

16 May 2024

Incidence of venous thromboembolism and bleeding in patients with malignant central nervous system neoplasm: systematic review and meta-analysis

PONE-D-23-39566R3

Dear Dr. Peres,

We’re pleased to inform you that your manuscript has been judged scientifically suitable for publication and will be formally accepted for publication once it meets all outstanding technical requirements.

Kind regards,

Omar A. Almohammed, Ph.D.

Academic Editor

PLOS ONE

---

## [Editor Report · Acceptance letter]

23 May 2024

PONE-D-23-39566R3 

PLOS ONE

Dear Dr. Peres, 

I'm pleased to inform you that your manuscript has been deemed suitable for publication in PLOS ONE. Congratulations! Your manuscript is now being handed over to our production team.

Kind regards, 

on behalf of

Dr. Omar A. Almohammed 

Academic Editor

PLOS ONE